# Characterization of Intrinsic Radiation Sensitivity in a Diverse Panel of Normal, Cancerous and CRISPR-Modified Cell Lines

**DOI:** 10.3390/ijms24097861

**Published:** 2023-04-26

**Authors:** Francisco D. C. Guerra Liberal, Stephen J. McMahon

**Affiliations:** The Patrick G Johnston Centre for Cancer Research, Queen’s University Belfast, Belfast BT7 1NN, UK; stephen.mcmahon@qub.ac.uk

**Keywords:** intrinsic radiosensitivity, predictive assay, radiation, cancer cells, DNA damage, CRISPR-Cas9

## Abstract

Intrinsic radiosensitivity is a major determinant of radiation response. Despite the extensive amount of radiobiological data available, variability among different studies makes it very difficult to produce high-quality radiosensitivity biomarkers or predictive models. Here, we characterize a panel of 27 human cell lines, including those derived from lung cancer, prostate cancer, and normal tissues. In addition, we used CRISPR-Cas9 to generate a panel of lines with known DNA repair defects. These cells were characterised by measuring a range of biological features, including the induction and repair of DNA double-strand breaks (DSBs), cell cycle distribution, ploidy, and clonogenic survival following X-ray irradiation. These results offer a robust dataset without inter-experimental variabilities for model development. In addition, we used these results to explore correlations between potential determinants of radiosensitivity. There was a wide variation in the intrinsic radiosensitivity of cell lines, with cell line Mean Inactivation Doses (MID) ranging from 1.3 to 3.4 Gy for cell lines, and as low as 0.65 Gy in Lig4^−/−^ cells. Similar substantial variability was seen in the other parameters, including baseline DNA damage, plating efficiency, and ploidy. In the CRISPR-modified cell lines, residual DSBs were good predictors of cell survival (R^2^ = 0.78, *p* = 0.009), as were induced levels of DSBs (R^2^ = 0.61, *p* = 0.01). However, amongst the normal and cancerous cells, none of the measured parameters correlated strongly with MID (R^2^ < 0.45), and the only metrics with statistically significant associations are plating efficiency (R^2^ = 0.31, *p* = 0.01) and percentage of cell in S phase (R^2^ = 0.37, *p* = 0.005). While these data provide a valuable dataset for the modelling of radiobiological responses, the differences in the predictive power of residual DSBs between CRISPR-modified and other subgroups suggest that genetic alterations in other pathways, such as proliferation and metabolism, may have a greater impact on cellular radiation response. These pathways are often neglected in response modelling and should be considered in the future.

## 1. Introduction

The intrinsic radiosensitivity of normal and tumour tissues is a major determinant of the outcome of any radiation-based treatment. A significant correlation between the intrinsic radiosensitivity of tumour cells, measured by in vitro clonogenic assays, and patient response to radiotherapy has previously been demonstrated [1,2]. The accurate prediction of radiation response and/or intrinsic radiosensitivity would be a major milestone towards effective personalised treatments.

Currently, radiotherapy, for a given cancer type, is typically given with a standard dose determined from clinical experience and population-level clinical trials. However, cancer is a highly heterogeneous disease, driven by numerous genetic alterations which significantly affect radiosensitivity. This genetic variability is reflected in estimates which suggest that cancers at the same site which receive the same treatment may vary in radiosensitivity by 25% or more [3,4]. As a result, a significant number of patients are certainly under- or over-dosed, reducing the clinical benefit of radiotherapy.

The clonogenic assay has been the gold standard method for measuring cellular radiosensitivity, providing a large amount of useful data. However, most of these studies were completed in different conditions and by different investigators, giving substantial variability in reported survival values, as elegantly demonstrated by Nuryadi et al. in the A549 cell model [5]. This same survival variability is observed in published data from other cell models, such as HT116, PC3, H460, DU145, SW480, HT29, and HT1299 (Figure 1) [6]. This enormous variability makes it very difficult to produce accurate radiosensitivity biomarkers or predictive models.

Furthermore, the clonogenic assay has limitations, including the extended time required for colony growth (typically 7 to 10 days or longer), and the low efficiency of clone formation in certain cell lines, which can make this assay unsuitable for such cells. This highlights the need for the application of new rapid predictive assays and models of radiation response if intrinsic sensitivity is to be translated as a clinical tool.

The other obvious candidate as a radiosensitivity marker is DNA double-strand breaks (DSBs). Their relationship to clonogenic survival has been previously studied, and some investigators found correlation between the two assays [7,8,9,10,11], but not all [12]. The weakness of these studies has been that they were performed in a small number of cell lines. Additionally, micronucleus and apoptosis assays have also been studied as end points for cellular radiosensitivity; however, there is no clear agreement regarding the predictive power of these metrics [11,13,14].

The aim of this study was to characterize radiosensitivity across a diverse panel of human cell lines. This combined clonogenic assays with measuring different biological features as surrogate markers of radiation response, including induction and repair of radiation-induced DSBs, cell cycle distribution, and ploidy. This was first performed in a controlled panel of cell lines with known DNA repair defects introduced using CRISPR-Cas9. This was then expanded into a more diverse panel of human cells with different genetic backgrounds. This enabled the evaluation of the predictive power of these baseline features, while at the same time beginning to build a robust dataset with reduced inter-experimental variabilities to support future modelling.

## 2. Results

### 2.1. X-Ray Radiosensitivity of CRISPR-Modified Cell Lines

Radiation survival curves for CRISPR-modified cell lines with known DNA repair defects are shown in Figure 2A. It is easily observed that removing a key protein from cells’ DNA damage response systems, such as LIG4, ATM, DNA-PK encoded by the PRKDC gene, Artemis encoded by the DCLRE1C gene, or BRCA1 drastically increases sensitivity when compared to wild-type cells. On the contrary, deleting p53 or BAX slightly increases cells’ resistance to radiation, and depletion of FANCD2 has a minimal impact in clonogenic survival after radiation exposure (Figure 2B).

Additionally, cellular DSB repair characteristics were examined by immunofluorescence staining of 53BP1 for various times after 2 Gy exposure. The DSB repair curves are shown in Figure 2C. Wild-type cells are capable of repairing most radiation-induced DSBs, with 91 ± 2% of DSBs repaired in 24 h. Similar repair percentages are seen with cells depleted of p53 (89 ± 3%), Artemis (89 ± 1%), BAX (93 ± 5%), and FANCD2 (94 ± 4%). However, removing key proteins of the Homologous Recombination (HR) and Non-homologous End Joining (NHEJ) repair pathways dramatically decreases cellular repair capabilities with repair percentages of 78 ± 3% for DNA-PK null cells, 73 ± 4 for BRCA1 null cells, 65 ± 5% for ATM null cells, and 50 ± 4% for LIG4 null cells. The depletion of key proteins also impacts the background level of DSBs in cells, as wild-type RPE-1 has almost no baseline DSBs, but cells depleted of BRCA1, p53 and DNA-PK, or FANCD2 present, on average, more than twice the level of background damage than wild-type cells, as seen in Figure 2D.

A summary of CRISPR-modified cell features and radiation response parameters, such as SF2 (fraction of cells surviving exposure to 2 Gy), D10% (radiation dose required to inactivate 90% of cell population), MID (mean inactivation dose), and plating efficiency, as well as distribution of cells in each phase of cell cycle, is given in Table 1.

### 2.2. Correlation between Cellular Features and Mean Inactivation Dose (MID) in CRISPR-Modified Cells

To identify if any cellular features provided an effective surrogate for radiosensitivity, we evaluated the correlation between MID and baseline DSBs (indirectly measured by the number of 53BP1 foci in control samples), radiation-induced DSBs (indirectly measured by the number of 53BP1 foci 30 min after irradiation), residual DSBs (indirectly measured by the number of 53BP1 foci 24 h after irradiation), cell cycle distribution and plating efficiency. The resulting correlation for each feature is plotted in Figure 3. Significant correlations were observed with induced DSBs (R^2^ = 0.61, *p* = 0.013), residual DSBs (R^2^ = 0.78, *p* = 0.009), and percentage of cell in S-phase of cell cycle (R^2^ = 0.46, *p* = 0.044). Plating efficiency is also statistically significantly correlated with MID (R^2^ = 0.45, *p* = 0.048), but this is dominated by the single outlying Lig4 KO cell line, and no significant trend is observed amongst other cell lines; however, and perhaps surprisingly, no significant correlation was seen between baseline DSBs and MID (R^2^ = 0.02, *p* = 0.729).

Moreover, it was observed that our survival data from the CRISPR-modified cell lines correlated well with Z-score values reported in an elegant CRISPR library study performed by Olivieri et al. (R^2^ = 0.78, *p* = 0.001) (Figure 3H) [15], suggesting good reproducibility between these two very different approaches for determining the impact of DNA repair defects on radiosensitivity.

### 2.3. X-Ray Radiosensitivity of Cancer and Normal Human Cells

We then performed the same evaluation of radiation response in a diverse panel of human cell lines, including normal lines, and cancerous lines originating from the prostate and lung. Figure 4 shows the radiation survival curves. Table 2 summarizes the cellular features of each cell line, such as ploidy, plating efficiency, SF2, and MID. There is also a wide variation in all the evaluated metrics among these cells, with A549, DU145, and PNT2 cell lines showing greatest resistance in their respective panels (lung cancer, prostate cancer, and normal). In contrast, H441, C4-2B, and MRC5 cell lines were the most sensitive cell lines in their respective panels (lung cancer, prostate cancer, and normal).

Significant variation was seen in sensitivity across all cell lines, with MID ranges of 1.5–3.4, 1.9–3.1, and 1.3–3.4 Gy, for lung, prostate, and normal, respectively. Notably, the range of sensitivity in supposedly ‘normal’ cells is the greatest among all the groups of cell lines considered here.

### 2.4. DNA DSB Repair Rates

Radiation-induced DSB repair curves are shown in Figure 5 for this cell line panel. The initial number of foci, baseline, and residual damage are shown in Table 2. It is possible to observe from the data that the background level of DSBs has an extensive range with some cell lines showing almost no baseline DSBs, such as RPE-1, A549, PNT2, and MRC5, while other cell lines show elevated levels of baseline DSBs, such as WPMY-1 and DU145, as shown in Figure 5D.

It is interesting to note that there is no statistically significant difference between the initial number of foci for the cell lines in the lung cancer and normal cell panels (*p* = 0.87 and *p* = 0.83, respectively, 1-way ANOVA), although this is almost significantly different (*p* = 0.053 1-way ANOVA) in the prostate cancer panel. Additionally, considering repair rates, there is a significant difference (*p* = 0.014) in the repair rates of cell lines in the lung cancer group. In contrast, prostate and normal cell lines have statistically similar repair rates within their respective groups (*p* = 0.55 and *p* = 0.38, respectively). The best fit repair rate for each of these two groups of cells are 0.21 ± 0.04 and 0.26 ± 0.04 h^−1^ for prostate and normal cells, respectively. Notably, unlike the CRISPR-modified cell lines, these lines all show a high degree of repair (95 ± 3%).

Yields of γH2AX foci were also obtained for a subset of cell lines to confirm the correspondence of these two markers. An excellent correlation was found between the number of 53BP1 foci and the number of γH2AX foci ($R^{2}$ = 0.99), with γH2AX foci systematically being approximately 14% higher in each condition and cell line (Appendix A).

### 2.5. Baseline Cell Cycle Profile and Ploidy

We measured the cell cycle distribution of each cell line by flow cytometry. The percentage of cells in each phase of the cell cycle ranged from 47.7% (PNT2) to 66.3% (LNCAP) for G1 phase, from 9.4% (AGO1522) to 15.1% (WPMY-1) for S phase, and from 19.1% (RWPE) to 39.9% (PNT2) for G2/M phase (Table 2). 

Our ploidy measurements were in excellent agreement with values reported by the Sanger Cell Model Passport (Appendix A, $R^{2}$ = 0.93, *p* < 0.0001). As expected, all normal cell lines have a ploidy value around 2, with exception of WPMY-1 that was immortalized with a SV40-large-T antigen gene and which underwent significant genome duplication. A wide range of values is observed in the cancer cell lines, from 2.12 for 22RV1 to 3.95 for PC-3. Interestingly, our data showed no statistically significant correlation between cell ploidy and the initial number of DSBs induced by 2 Gy ($R^{2}$ = 0.06 and *p* = 0.31) (Figure 6), despite a linear relationship being proposed by many models.

### 2.6. Correlation between Cellular Features and Mean Inactivation Dose (MID)

To identify if any radiobiological parameters provided an effective surrogate for survival in this more complex and diverse panel of cells, we evaluated the correlation between MID and ploidy, baseline DSBs, radiation-induced DSBs, residual DSBs, and cell cycle distribution, as well as plating efficiency. The resulting correlation heatmap is plotted in Figure 7. Correlations were evaluated for each cell line subgroup, as well as for all cell lines together, excluding CRISPR-modified lines.

Strikingly, DNA damage parameters performed very poorly across these cell lines. While trends were seen in some subgroups (e.g., baseline DSBs in lung, or induced DSBs in prostate cancer lines), no significant correlation was observed between any DNA damage endpoint and MID. Indeed, normal and prostate cancer cell lines showed particularly poor correlations between cellular features and survival, with no metric showing significant predictive power. Some significant correlations were observed in the lung cancer subgroup with some other features, including the percentage of cells in G2 phase of the cell cycle (R^2^ = 0.72, *p* = 0.031), and plating efficiency (R^2^ = 0.79, *p* = 0.018) (Appendix A).

Unfortunately, most of these metrics perform poorly in the population as a whole. Only plating efficiency (R^2^ = 0.31, *p* = 0.013) and percentage of cells in S phase (R^2^ = 0.37, *p* = 0.005) showed statistically significant predictive power, although correlation coefficients were relatively low. Together, these observations suggest that while DSBs do play a significant role in radiation response, in a diverse panel of genetically heterogeneous cell lines they are not individually strong predictors of response.

## 3. Discussion

It is widely accepted that there is a vast variability in intrinsic radiosensitivity between different cancer cells which is not currently accounted for. As a result, the identification of a biological endpoint predictive of tumour radiosensitivity would make an important contribution to enhancing the effectiveness of radiotherapy, by allowing treatments to be planned at an individual patient level.

This heterogeneity in radiosensitivity was seen in the present study, with SF2 values from 0.041 to 0.71 (MID from 0.70 to 3.36 Gy). This is in agreement with previous reports of radiosensitivity across a wide variety of human tumours by Amundson et al. (SF2 range from 0.038 to 0.95) [16]. The lower range of SF2 reported in Amundson’s study was mostly seen in lymphoid tumours, which were not evaluated in our panel. By contrast, in our data, the only cell line that had extreme sensitivity was RPE-1 LIG4^−/−^, and the second most sensitive cell line in our panel was RPE-1 ATM^−/−^. ATM and LIG4 are major players in cellular DNA damage response and DNA repair, and their loss is well-established to be associated with radiation hypersensitivity [15].

Indeed, our findings show remarkable concordance with previously published results from a CRISPR screen studying radiation response in RPE-1 p53^−/−^ cells, despite employing different methods to assess the impact of DNA repair deficiencies on radiosensitivity. Specifically, while we investigated clonogenic survival, the screen study quantified raw numbers of cells with the loss of particular genes [15]. This not only underscores the reproducibility of different approaches, but also increases confidence in the robustness of our data presented here. We observed a near-perfect correlation among various knockout models, with the exception of BRCA1 null cells (R^2^ = 0.98, *p* = 0.002). This discrepancy in BRCA1 data could potentially be attributed to the limitations of CRISPR dropout screens in genes that are essential for cell proliferation, as well as genes with multiple cellular functions, as has been previously reported in BRCA1 [17]. Nonetheless, despite this discrepancy, our results align closely with previously reported data and strengthen the overall reliability of our findings.

The determination of radiosensitivity by the clonogenic assay is a lengthy process, so there is a need to understand cellular mechanisms and parameters that might enable the prediction of intrinsic radiosensitivity. The development of such rapid techniques would enable them to inform clinical decision-making and allow personalized treatment planning.

Here, we evaluated the relationship of cellular radiosensitivity with different cellular characteristics in a two-step approach; first in a controlled environment where the only difference between cell models is targeted mutations in specific DNA damage repair genes, then in diverse panel of human cell lines where cells have a more complex genetic profile. A wide variation was observed in cellular features, such as ploidy, plating efficiency, and DNA damage repair rates, among the cells used in this study.

Radiation-induced DSBs and their repair is attractive as a clinical and pre-clinical test, as results can be available in a few hours. Moreover, DSB repair is an obvious candidate marker for radiation sensitivity because DSBs are considered the most critical type of DNA damage caused by radiation. The relationship between residual levels of DSBs or repair rates to cell survival in cancer has been a topic of numerous investigations [7,8,9,10,11,12]. In CRISPR-modified cells with DNA repair defects, the number of residual DSBs is the metric which best predicts cell survival ($R^{2}$ = 0.78, *p* = 0.009). This excellent performance in the CRISPR-modified cell lines is likely due to the very controlled environment where the only difference between cell models is one mutation in a specific DNA damage repair gene.

On the contrary, in the whole panel of human cell lines, where cells have a more complex genetic profile, residual DSBs fail to predict survival ($R^{2}$ = 0.02, *p* = 0.542), as do other metrics associated with DSBs, such as baseline levels of damage and levels of induced damage. This is despite many of the cell lines we investigated having mutations in DSB repair pathways (Appendix A). Interestingly, even among the subgroup of normal human cell lines that are assumed to have an intact genetic background, none of the analysed metrics demonstrate predictive power, despite significant variation in sensitivity. It is important to note that most of these normal cell lines were immortalized and the impact of immortalization on radiation response is complex and context-dependent, and further research is needed to fully understand the underlying mechanisms. Tissue type may also play a role in these differences, as the more varied backgrounds of these normal lines may impact on involved proliferative, metabolic, and cell death processes.

However, while CRISPR knockout generates severe homozygous loss of function mutations, many of these mutations have less clear significance or are heterozygous, potentially having a reduced impact on DSB repair. This is supported by more general mutational data, such as in the Cancer Cell Line Encyclopedia [18], where 22% of cell lines have at least one damaging mutation in the NHEJ or HR pathways, but only 2% of cell lines have a homozygous damaging mutation of the kind assessed in most radiobiological DNA repair studies. This suggests that, in more diverse cell lines, DSBs alone may not be as strong a predictor of response.

Considering other possible parameters, in this study we did not find a correlation between survival and ploidy. This agrees with the study of West et al., where no statistically significant differences in SF2 values could be found between diploid and aneuploid cervical carcinomas [19].

Cellular sensitivity is known to vary during different phases of the cell cycle. Typically, cells are most sensitive to irradiation during mitosis and in late G2, less sensitive in G1, and least sensitive during the latter part of S phase [20]. Thus, cell cycle profile distribution is another possible predictor of radiosensitivity. Nonetheless, there is some controversy around the impact of cell cycle distribution in radiosensitivity with some studies showing correlations while others have not [13,21]. Here, we found suggestions of correlation between intrinsic sensitivity and cell cycle distribution, particularly with the percentage of cell in S phase in both groups of cells, CRISPR-modified ($R^{2}$ = 0.46, *p* = 0.044) and pooled cell lines ($R^{2}$ = 0.37, *p* = 0.005).

Interestingly, although S-phase is typically suggested to be a more resistant phase of the cell cycle, in the CRISPR-modified cell lines this correlation is negative (Figure 3E). This suggests that this may be an indirect effect, resulting from, e.g., slowed progression through S phase when DNA repair defects are present. A positive correlation is seen as expected between S-phase population and MID in the pooled cell lines. Somewhat surprising is the substantial correlation of percentage of cell in G2 phase of the cycle in the subgroup of lung cancer cell lines ($R^{2}$ = 0.72, *p* = 0.032), as this metric performed poorly for all the other subgroups ($R^{2}$ < 0.35, *p* > 0.12).

Similar behaviour was observed with plating efficiency, where it had some predictive power for survival ($R^{2}$ = 0.31, *p* = 0.013) across the pooled cell lines. This metric performed well in the subgroup of lung cancer cell lines ($R^{2}$ = 0.79, *p* = 0.018), but poorly for the prostate cancer cells ($R^{2}$ = 0.03 *p* = 0.77). The lack of correlation between plating efficiency and survival has also been previously reported [22]. Plating efficiency had a similar predictive power on the CRISPR cell lines subgroup ($R^{2}$ = 0.45, *p* = 0.048), although this is dominated by the LIG4 knockout line. Taken together, we can conclude that plating efficiency may give some indications of sensitivity but is not a generally reliable predictive feature alone.

Moving beyond individual correlations, these data may be useful, in combination with other resources, to help develop new mechanistic models of radiation response which can incorporate these different parameters together in predictive tools [23]. To support this, results from this paper are made available in full in a Appendix A.

For future model development, the variability between and within each of the subgroups, combined with the lack of correlation with DNA damage metrics, leads us to hypothesize that genetic alterations in proliferation and metabolic pathways may have a significant impact in cellular radiation response, as cellular metabolism, proliferation, and antioxidant pathways play crucial roles in the response of cells to radiation.

As an example, high expression of the mTOR protein, an important protein in the regulation of cell proliferation and metabolism, was found to be associated with poor prognosis in cervical cancer treated with radiotherapy [24]. Therefore, pathways such as PI3K/Akt/mTOR and MAPK/ERK have an important role in intrinsic radiosensitivity and clinical radiation response and should be investigated.

Furthermore, metabolic pathways also control the production of energy and essential molecules needed for cell survival and repair. Understanding how radiation affects cellular metabolism can provide insights into the mechanisms of radiation response and may lead to strategies for enhancing radiation efficacy. The disruption of cellular metabolism with mannose has been shown to result in impaired cancer cell growth and survival [25]. This effect is thought to be mediated through multiple mechanisms, including the disruption of glycolysis, leading to a decreased energy production, modulation of signalling pathways involved in cancer progression, such as the PI3K/Akt pathway and enhancement of anti-cancer immune response [26].

Similarly, radiation exposure increases the levels of reactive oxygen species (ROS) and other damaging molecules, overwhelming the cellular antioxidant defence mechanisms. Subsequently, this can cause oxidative stress, further DNA damage and cell death. Antioxidant pathways, such as the glutathione system, superoxide dismutase, and catalase, help to neutralize ROS and prevent oxidative damage [27]. Modulating these antioxidant pathways may enhance the cellular antioxidant capacity and reduce radiation-induced damage [28].

In summary a better understanding of the interplay between cellular metabolism, proliferation, antioxidant pathways, and radiation response is essential for elucidating the mechanisms of radiation response and may lead to the development of strategies to increase radiation efficacy in cancer treatments.

## 4. Materials and Methods

### 4.1. Cell Lines

In this study a total of 27 cell lines were used, divided in four groups: (1) 6 Human lung cancer cell lines acquired from ATCC (H460, A549, H1792, SW1573, H23, and H441); (2) 5 Human prostate cancer cell lines acquired from ATCC (PC-3, DU145, 22RV1, C42B, and LNCAP); (3) 8 Human normal cell lines acquired from ATCC or ECACC (PNT1A, MRC5, RPE-1, WPMY1, AGO1522, RWPE1, MCF10A, and PNT2); and (4) 8 CRISPR-modified cell lines, derived from the RPE-1 cell line with stable knock-out of p53, ATM, DCLRE1C (Artemis), BRCA1, BAX, LIG4, FANCD2, and PRKDC (DNA-PKcs).

All cell lines were maintained in supplier recommended complete medium (Appendix A). All cultures were incubated at humidified 37 °C in 5% CO_2_.

The RPE-1 cells were selected for CRISPR-Cas9 studies due to their stable epithelial cell characteristics, with minimal genetic modifications, which closely mimic normal tissue biology. Additionally, its widespread usage in previous CRISPR studies provides a robust foundation of data for comparison, confirming its suitability as a reliable and well-established cellular model.

The process of selecting tumoral and normal human cell lines in this study was carried out based on factors such as cell availability, previously described levels of radiosensitivity and their widespread use in radiation studies. Lung and prostate cell lines were selected in this initial comparison as radiotherapy is frequently used as a primary treatment in these cancers.

### 4.2. Establishment of DNA Repair Deficient CRISPR/Cas9 Stable Cell Lines

CRISPR-Cas9 modified cell lines were generated using a transient transfection of CRISPR-Cas9 in RPE-1 cells, by delivering a ribonucleoprotein (RNP) complex to recombinant Cas9 coupled to a specific guide RNA following manufacturer’s instructions. In brief, an RNP complex was assembled by mixing the tracrRNA (100 µM)(#1072532 IDT, Coralville, AI, USA) and crRNA (100 µM) (predesigned crRNA were obtained from IDT, Coralville, AI, USA) in equimolar concentrations and Cas9 Nuclease V3 (61 µM) (#1081058 IDT, Coralville, AI, USA). A list of crRNA used in this study can be found in Appendix A.

Delivery of the RNP complex to cells was performed via Lonza nucleofection. For that, 3.5 × 10^5^ cells suspended in 20 µL of Nucleofector Solution (P3 Primary Cell 4D-NucleofectorTM X Kit S, #V4XP-3032, Lonza, Basel, Switzerland) supplemented with 5 µL of previously assembled RNP complex and 1 µL of IDT electroporation enhancer (100 µM) (#1075916 IDT, Coralville, AI, USA) were used for each nucleofection. After nucleofection cells were expanded for clonal selection.

All cell lines were generated as single knockouts, apart from BRCA1 which was combined with P53 knockout, as cells with only BRCA1 knockout were not viable.

The RPE-1 cells were selected for CRISPR-Cas9 studies due to their stable epithelial cell characteristics, with minimal genetic modifications, which closely mimic normal tissue biology. Additionally, its widespread usage in previous CRISPR studies provides a robust foundation of data for comparison, confirming its suitability as a reliable and well-established cellular model.

### 4.3. Knock-Out Validation by Western Blot

Stable knock-out was validated by Western Blot of the protein of interest in populations derived from single clones (Appendix A). Cells were lysed in RIPA buffer (150 mM Tris-HCl pH 8; 50 mM NaCl; 1% *v*/*v* NP-40, 0.5% *v*/*v* sodium deoxycholate (10%) and 0.1% *v*/*v* SDS (10%)) supplemented with protease inhibitor (cOmplete Mini, Roche, Basel, Switzerland).

A 40 µg total protein sample was loaded onto an SDS-PAGE gel, and, after electrophoresis, the proteins were blotted on a nitrocellulose membrane (Life Technologies, Carlsbad, CA, USA). The membranes were blocked with 5% non-fat milk in PBS-Tween (0.1%) and incubated overnight at 4 °C with primary antibody diluted in milk (Appendix A). After washing with PBS-Tween, membranes were incubated in their secondary anti-rabbit and anti-mouse horseradish peroxidase-conjugated antibodies diluted at 1:2000 at room temperature for 1 h. The membranes were then washed and developed with Luminata Crescendo Western Blot HRP substrate (Milipore, Burlington, MA, USA) using the GBox Imager by Synagene (Synagene UK, Cambridge, UK).

### 4.4. Clonogenic Survival Assay

The colony formation assay was carried out according to published methods [29]. Cells were seeded into six-well plates with an optimal cell density according to the absorbed dose. On the following day, cells were irradiated with doses of 0 to 8 Gy of X-rays, at a dose rate of 0.59 Gy/min using a 225 kVp, 13.3 mA X-RAD 225 radiation source (Precision X-ray Inc., Madison, CT, USA).

After irradiation, cells were incubated for 7 to 11 days depending on the cell line. The colonies were stained with 4% crystal violet solution in ethanol and were manually counted, with a colony defined as consisting of at least 50 cells. From these counts, plating efficiency (PE) and survival fraction (SF) were calculated. Survival fraction was defined as the number of colonies formed after irradiation divided by the number of cells seeded, corrected for the PE of unirradiated cells. Data were fit to the linear quadratic equation ($SF=e^{-\left(\alpha D+\beta D^{2}\right)}$) using non-linear regression weighted for standard deviation. The Mean Inactivation Dose (MID) is defined as the area under the dose–response curve.

### 4.5. DNA Damage Immunofluorescence Assay

Following 2 Gy irradiation, cells were fixed in a 50:50 methanol–acetone solution and permeabilized (0.5% Triton X-100 in PBS) at predetermined time points before being blocked in blocking buffer (5% FBS an 0.1% Triton X-100 in PBS) and stained with 53BP1 primary antibody (1:5000) (#NB100-304, Novus Biologicals, Centennial, CO, USA) and γH2AX primary antibody (1:10000) (#05-636-I, Merks Chemicals, Frankfurt, Germany) for 1 h before being washed three times and stained with Alexa Flour 568 goat anti-rabbit IgG secondary antibody (#A21429, Life Technologies, Carlsbad, CA, USA) and Alexa Flour 488 goat anti-mouse IgG secondary antibody (#A21131, Life Technologies, Carlsbad, CA, USA) (1:2000) in the dark for 1 h. Following staining, cells were washed three times and mounted onto microscope slides using Prolong Gold anti-fade reagent with DAPI (#P36930, Invitrogen, Carlsbad, CA, USA). Foci were manually counted from the whole nucleus of 50 randomly selected cells on each sample with a Nikon Eclipse Ti microscope (Nikon Corporation, Tokyo, Japan), using a 60× objective, representative images of 53BP1 foci can be found in Appendix A. Data are presented as the mean values of foci per cell and the respective standard deviation of three independent experiments. Data presented here are corrected for control samples by subtracting off the number of foci in unirradiated cells. For repair kinetic analysis, foci data were then fit in GraphPad Prism 9 using non-linear regression to an exponential decay, $N=(N_{0}-plateau)*e^{-kt}+plateau$, where $N_{0}$ represents the initial number of foci, plateau represents the residual damage and k is the rate of DSB repair.

### 4.6. Cell Cycle Profile and Ploidy Analysis

Exponentially growing cells were harvested before being fixed in 100% ice-cold ethanol and left at 4 °C overnight. Samples were then centrifuged, the excess ethanol was removed, the cell pellets were then resuspended in PBS and centrifuged again before being resuspended in 500 µL of PI/RNAse A. The samples were incubated at 37 °C for 30 min before being analysed in a BD Accuri C6 Plus Flow Cytometer (BD Biosciences, Franklin Lakes, NJ, USA). In total, 10,000 flow cytometer events were collected and analysed per sample. Quantification was carried out using the BD Accuri C6 Plus Analysis software. To perform ploidy analysis, the value of each sample’s G1 median fluorescence intensity (MFI) peak was then compared to the value of the MFI peak of Chicken Erythrocyte Nuclei (CEN) Singlets (#1013, BioSure, Grass Valley, CA, USA). The ratio of the G1 MFI peak value to the CEN singlets MFI peak value was then compared with reported ploidy value of cells by Sanger (https://cellmodelpassports.sanger.ac.uk (accessed on 8 November 2021)) where available, showing an $R^{2}$ of 0.93 (Appendix A).

### 4.7. Statistical Analysis

All experiments were performed in triplicate, data are presented as mean values and respective standard deviation. Unpaired Student’s *t*-test and one-way ANOVA were used for statistical evaluation as appropriate. All statistics and graph plotting used GraphPad Prism 9.0 (GraphPad, Boston, MA, USA).

## 5. Conclusions

Our results indicate that this diverse panel of human cell lines have a wide variation in radiosensitivity and cellular characteristics. Residual radiation-induced DSBs were clearly related to intrinsic radiosensitivity in the subgroup of CRISPR-modified cells with well-defined DNA repair mutations, but they failed to predict survival in a group of cells with a more complex genetic profile. These data suggest that purely DNA damage-based analyses have a limited predictive power in diverse cell lines and supports further investigation into different metabolic and proliferative cellular pathways and their relationship with radiation response. Additionally, these data may assist the development of new predictive models of radiation response.

## Figures and Tables

**Figure 1 ijms-24-07861-f001:**
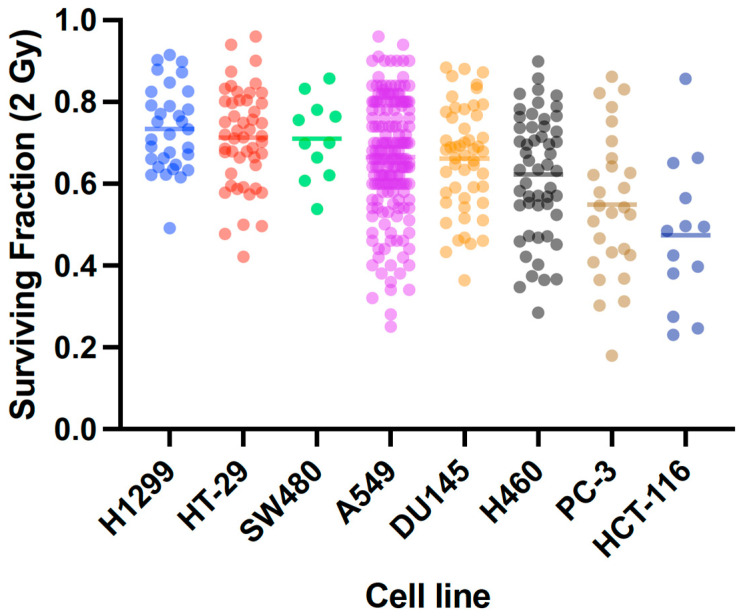
Variation in reported Surviving Fraction following 2 Gy irradiation (SF2) across a panel of cell lines, extracted from a literature survey of independent studies. Points are individual measurements; horizontal lines denote mean reported survival. In all cases, very large variation can be seen, with significant overlap between all cell lines.

**Figure 2 ijms-24-07861-f002:**
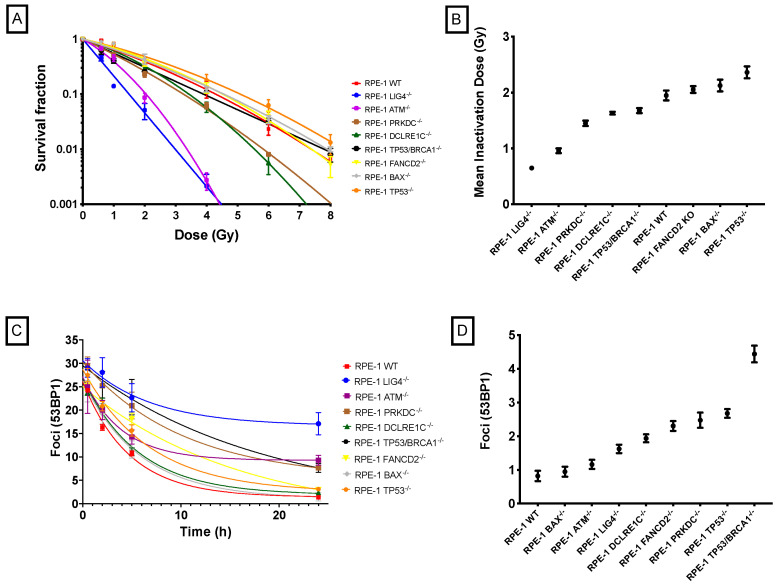
Radiosensitivity of CRISPR-modified cell lines with known defects in DNA repair pathways: (**A**) radiation-induced clonogenic survival curves, (**B**) mean inactivation values (MID) for each CRISPR cell model, (**C**) radiation-induced DSBs repair curves, and (**D**) baseline levels of DNA damage for each CRISPR cell model.

**Figure 3 ijms-24-07861-f003:**
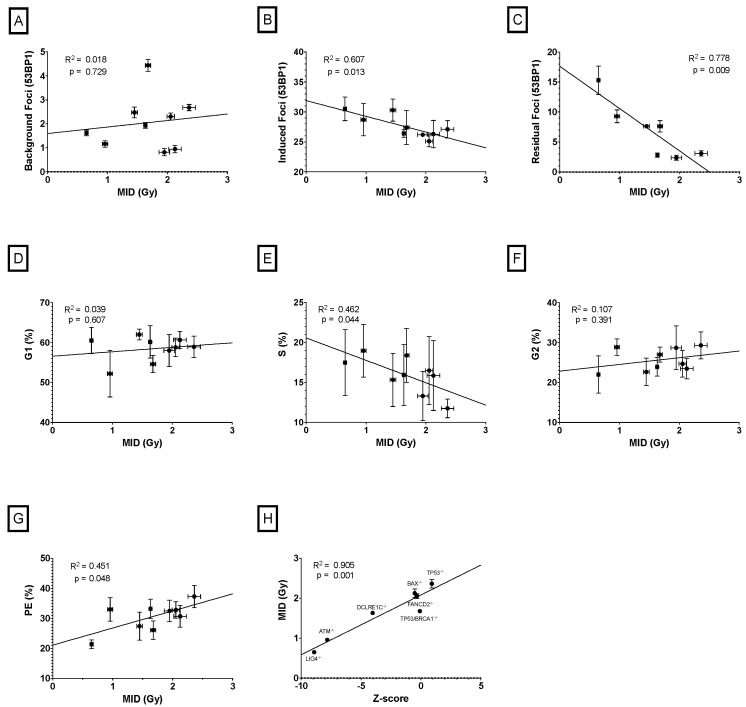
Evaluation of the predictive power of different cellular features as an indirect measurement of survival. Correlation between MID and (**A**) baseline DSBs, (**B**) radiation-induced DSBs, (**C**) residual DSBs after 24 h post radiation, (**D**) percentage of cells in G1 phase of cell cycle, (**E**) percentage of cells in S phase of cell cycle, (**F**) percentage of cells in G2/M phase of cell cycle, and (**G**) plating efficiency. (**H**) Correlation between MID reported for each KO cell line in this study with radiosensitivity Z-score values previously published from a CRISPR library screen in RPE-1 p53^−/−^ cells treated with radiation. Points represent the mean of three independent repeats and error bars indicate the standard deviation of the means.

**Figure 4 ijms-24-07861-f004:**
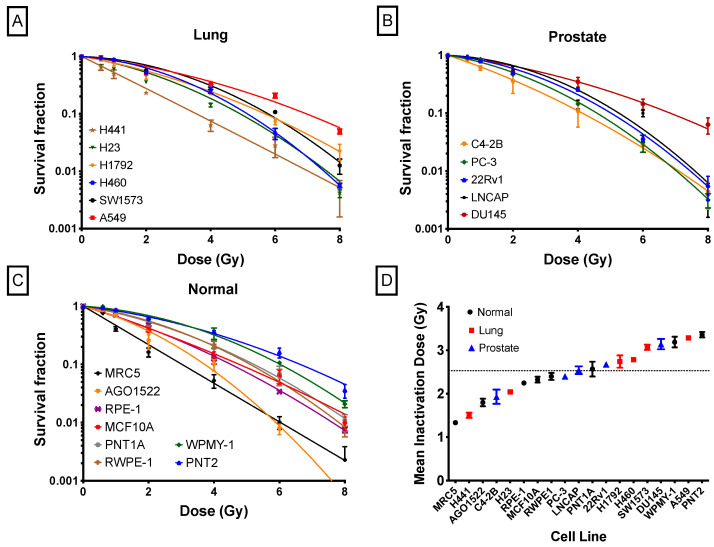
Radiation-induced survival curves: (**A**) lung cancer cell lines, (**B**) prostate cancer cell lines, (**C**) normal human cell lines, and (**D**) mean inactivation doses for each cell line. The dashed line represents the mean value across all cells. Experiments were carried out three times and error bars indicate the standard deviation of the means.

**Figure 5 ijms-24-07861-f005:**
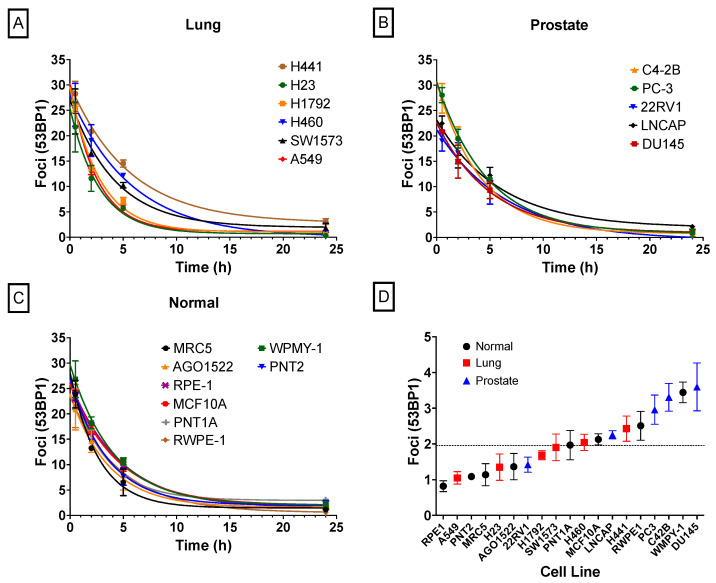
Repair kinetics of radiation-induced foci, measured by immunofluorescence of 53BP1 at 30 min, 2 h, 5 h, and 24 h after exposure to 2 Gy of radiation, divided into subgroups. (**A**) Lung cancer cell lines, (**B**) prostate cancer cell lines, (**C**) normal human cell lines, and (**D**) levels of background damage in control samples for each cell line. The dashed line represents the mean across all cells. Experiments were carried out three times and error bars indicate the standard deviation of the means.

**Figure 6 ijms-24-07861-f006:**
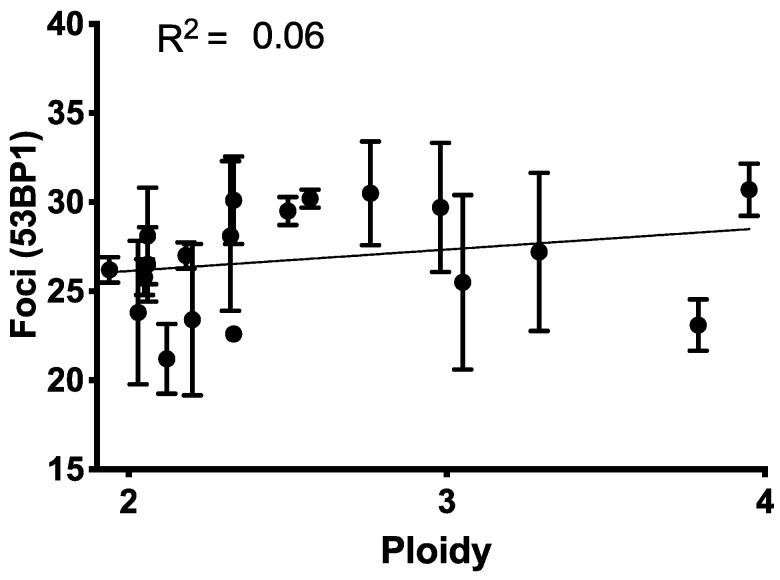
Initial yields of 53BP1 foci compared with cell ploidy. Although many models assume a linear relationship between these quantities, no statistically significant correlation is seen (*p* = 0.31).

**Figure 7 ijms-24-07861-f007:**
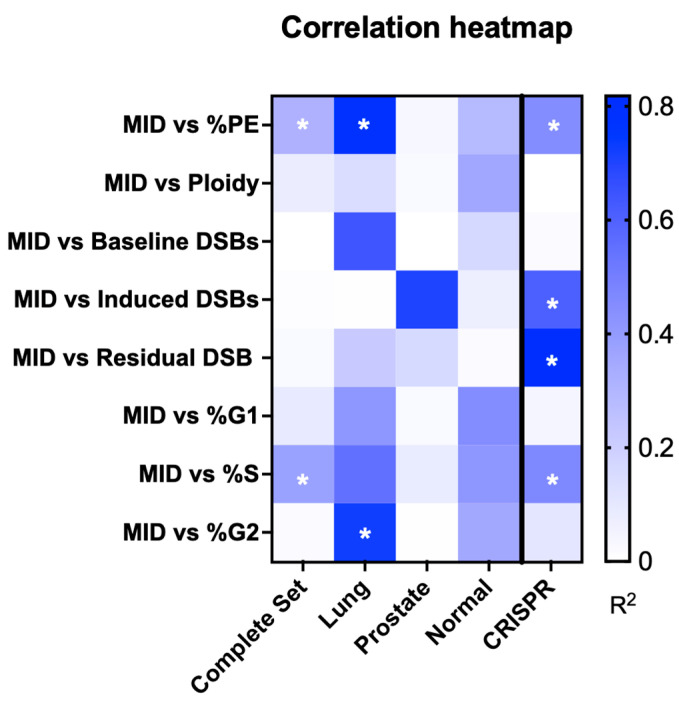
Heatmap of the potential predictive power of different cellular characteristics as a surrogate metric of radiation response, via correlation with MID. Residual damage after irradiation performed well as a sensitivity biomarker for CRISPR-modified cell lines, but failed to predict survival of cancer cells, likely due to their more complex genetic profile where alterations in other pathways, such as proliferation or metabolism, may have a larger impact on their radiation response. * Shows metrics which statistically significantly correlated with MID (*p* < 0.05).

**Table 1 ijms-24-07861-t001:** Characteristics and radiation survival of CRISPR-modified cell lines. PE—plating efficiency; SF2—survival fraction at 2 Gy; MID—mean inactivation dose; baseline DSBs—number of foci in control samples; induced DSBs—number of foci 30 min after 2 Gy exposure corrected for control; and residual DSBs—number of foci 24 h after 2 Gy exposure corrected for control. Standard deviation of the means for each parameter can be found in Appendix A.

Cell Line	Ploidy	Cell-Cycle (%)			PE (%)	SF2	MID	Baseline DSBs	Induced DSBs	Residual DSBs
		G1	S	G2/M						
RPE-1	1.94	58.0	13.3	28.7	32.5	0.39	1.95	0.9	26.2	2.4
RPE-1 LIG4^−/−^	1.94	60.5	17.5	22.0	21.4	0.05	0.65	1.7	30.5	15.3
RPE-1 ATM^−/−^	1.94	52.2	19.0	28.8	33.0	0.11	0.96	1.2	28.7	9.3
RPE-1 PRKDC^−/−^	1.94	62.0	15.3	22.7	27.4	0.26	1.45	2.5	30.3	7.6
RPE-1 DCLRE1C^−/−^	1.94	60.2	15.9	23.9	33.2	0.33	1.63	1.9	26.4	2.8
RPE-1 TP53/BRCA1^−/−^	1.94	54.6	18.4	27.0	26.1	0.31	1.67	4.4	27.4	7.6
RPE-1 FANCD2^−/−^	1.94	58.8	16.5	24.7	32.8	0.40	2.02	2.3	25.1	2.9
RPE-1 BAX^−/−^	1.94	60.7	15.8	23.5	30.7	0.41	2.05	0.9	26.3	1.9
RPE-1 TP53^−/−^	1.94	58.9	11.7	29.3	37.3	0.48	2.36	2.7	27.1	3.1

**Table 2 ijms-24-07861-t002:** Characteristics and radiation survival of cell lines used in this study. PE—plating efficiency; SF2—survival fraction at 2 Gy; MID—mean inactivation dose; baseline DSBs—number of foci in control samples; induced DSBs—number of foci 30 min after 2 Gy exposure corrected for control; and residual DSBs—number of foci 24 h after 2 Gy exposure corrected for control. Standard deviation of the means for each parameter can be found in Appendix A.

Cell Line	Ploidy	Cell-Cycle (%)			PE (%)	SF2	MID	Baseline DSBs	Induced DSBs	Residual DSBs
		G1	S	G2/M						
*H441*	2.33	57.1	9.1	33.8	10.0	0.27	1.54	6.3	30.1	2.8
*H23*	3.05	59.0	11.1	29.9	26.1	0.53	2.44	2.1	25.5	0.7
*H1792*	2.50	59.5	13.7	26.8	34.9	0.56	2.72	2.9	29.5	1.1
*H460*	2.32	64.4	12.5	23.1	65.3	0.63	2.79	3.8	28.1	0.1
*SW1573*	3.29	59.4	14.8	25.8	58.4	0.71	3.21	3.3	27.2	1.9
*A549*	2.57	62.9	11.7	25.4	63.3	0.67	3.40	1.2	30.2	1.2
*C42B*	2.76	64.3	10.1	25.6	9.9	0.37	1.88	6.0	30.5	0.8
*PC-3*	3.95	62.4	13.9	23.7	46.3	0.51	2.31	5.6	30.7	0.8
*22RV1*	2.12	48.5	13.5	38.0	45.1	0.59	2.65	2.5	21.2	0.5
*LNCAP*	3.79	66.3	10.5	23.2	4.3	0.67	2.93	5.8	23.1	1.9
*DU145*	2.33	60.6	13.2	26.2	40.4	0.65	3.12	6.7	22.6	0.9
*MRC5*	2.06	63.0	10.3	26.7	13.9	0.22	1.31	1.7	28.1	1.5
*AGO1522*	2.03	65.6	9.4	25.0	27.5	0.37	1.80	2.3	23.8	1.6
*RPE-1*	1.94	58.0	13.3	28.7	32.5	0.39	1.95	0.9	26.2	2.3
*MCF10A*	2.05	58.7	10.2	31.1	11.6	0.42	2.27	5.4	25.8	2.0
*PNT1A*	2.06	62.2	12.1	25.7	30.5	0.54	2.55	3.6	26.5	2.9
*RWPE1*	2.20	69.2	11.7	19.1	20.6	0.56	2.57	5.0	23.4	0.6
*WPMY-1*	2.98	49.2	15.1	35.7	20.8	0.69	3.22	6.3	29.7	2.0
*PNT2*	2.18	47.7	12.4	39.9	44.4	0.65	3.43	1.5	27.0	1.9

## Data Availability

All data reported for the cell lines in this paper has been made available as a supplementary datasheet for analysis and reuse.

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
