# Peer review of "Characterization of Intrinsic Radiation Sensitivity in a Diverse Panel of Normal, Cancerous and CRISPR-Modified Cell Lines"

_ijms, 2023, doi:10.3390/ijms24097861_

Round 1
Reviewer 1 Report
In their manuscript, the authors address a long-standing unresolved question of identifying markers that could predict the sensitivity of cancer cells to radiation. The interest of this study is to monitore physiological parameters such as plating efficiency, cell cycle, DNA repair (through 53BP1 and gH2AX foci at different times) alongside clonogenic survival. Interestingly, the first set of cells isogenic except for specific genes modified by CRISPR shows a strong correlation between DNA repair (in particular residual damage) and survival to irradiation. However, this correlation is lost in the set of cancer cells that differ in many genes and probably exhibit more complex defects in the different repair pathways and altered regulation of proliferation and metabolism.
Though it fails to provide new radiosensibility markers, this manuscript deserves to be published for its quality of its results and the clear demonstration that prediction of radiation response will require taking into account the full complexity of the cancer cells, i.e. the combination of many parameters.
Main comment:
In a general way, taking in account the topic of this manuscript there should be a more accurate description of the cell lines used and their characteristics.
The choice of the cells should be argumented. In particular, why choosing RPE1 for the CRISPR constructions? How the cancer cells were selected?
Normal cell lines are usually immortalized for such studies. Please precise the modification used in the list of the cell lines. Comment the possible impact of such modifications on radiosensitivity.
Why is the BRCA1 construction also TP53-/- ?
Minor comment:
Figure 1 : The points and values do not appear on the figure
Line 62 :comment, in the list of the disadvantages of clonogenic assay, do not forgot to cite the fact that some cell lines have a very poor efficiency of clone formation that prevents their use for radiosensitivity measurement.
Figure 3 H : These results should be commented. Number of cell line in common, differences between survival measurements methods etc…
Figure 7 : the color scale is entirely white!
Table 1 : there is an error in the cell cycle counts of RPE1-DCLRE1C (>100%). This cell line is not commented in the results
Results §2 line 119-124 : the use of DSB is inexact and should be defined as it is 53BP1 foci that were measured and not directly DSBs. Similarly the terms of residual DSBs and induced DSBs should be more precisely described (53BP1 foci at 24h and 1h?).
Figure 5. Since residual DSBs (53 BP1foci at 24h) seem to be the marlker with the best correlation to survival in normal cells why are the induced DSBs presented and discussed and not residual DSBs? Precise the time of the 53BP1 measurement in the legend of the figure
Author Response
Reviewer 1:
Reviewer: “In their manuscript, the authors address a long-standing unresolved question of identifying markers that could predict the sensitivity of cancer cells to radiation. The interest of this study is to monitore physiological parameters such as plating efficiency, cell cycle, DNA repair (through 53BP1 and gH2AX foci at different times) alongside clonogenic survival. Interestingly, the first set of cells isogenic except for specific genes modified by CRISPR shows a strong correlation between DNA repair (in particular residual damage) and survival to irradiation. However, this correlation is lost in the set of cancer cells that differ in many genes and probably exhibit more complex defects in the different repair pathways and altered regulation of proliferation and metabolism. Though it fails to provide new radiosensibility markers, this manuscript deserves to be published for its quality of its results and the clear demonstration that prediction of radiation response will require taking into account the full complexity of the cancer cells, i.e. the combination of many parameters.”
Authors: We sincerely thank the reviewer for their thoughtful evaluation of our manuscript. We appreciate their recognition of the quality of our results and the clear demonstration of the challenges in identifying radiosensitivity markers in cancer cells. Despite the lack of new radiosensitivity markers, we believe our manuscript provides valuable insights into the multifactorial nature of radiosensitivity in cancer cells, and we are grateful for Reviewer’s constructive comments that have helped improve our paper.
Main comment:
Reviewer: “In a general way, taking in account the topic of this manuscript there should be a more accurate description of the cell lines used and their characteristics.”
Authors: In this revised version of the manuscript, the authors have made substantial changes to the content of supplementary table 5. The updated version contains information regarding origin of cell lines, age of the patient at the time of sample collection, cell ploidy, karyotype, morphology, average doubling time, culture media, and culture media supplements. We would be happy to include any other factors the reviewer thinks are relevant here.
Reviewer: “The choice of the cells should be argumented. In particular, why choosing RPE1 for the CRISPR constructions? How the cancer cells were selected?”
Authors: Following the Reviewer’s recommendation we have now justified the selection of RPE-1 for CRISPR constructions and cell selection process as the following: “The RPE-1 cells were selected for CRISPR-Cas9 studies due to their stable epithelial cell characteristics, with minimal genetic modifications, which closely mimic normal tissue biology. Additionally, its widespread usage in previous CRISPR studies provides a robust foundation of data for comparison, confirming its suitability as a reliable and well-established cellular model. The process of selecting tumoral and normal human cell lines in this study was carried out based on factors such as cell availability, previously described levels of radiosensitivity and their widespread use in radiation studies. Lung and prostate cell lines were selected in this initial comparison as radiotherapy is frequently used as a primary treatment in these cancers.” This can be found in line 385 to 394 of the manuscript.
Reviewer: “Normal cell lines are usually immortalized for such studies. Please precise the modification used in the list of the cell lines. Comment the possible impact of such modifications on radiosensitivity.”
Authors: Following the Reviewer’s recommendation we have now introduced information regarding immortalization in supplementary table 5. Additionally, a short commentary of possible impacts of radiation response due to immortalization techniques was added as the following: “Interestingly, even among the subgroup of normal human cell lines that are assumed to have an intact genetic background, none of the analyzed metrics demonstrate predictive power, despite significant variation in sensitivity. It's important to note that most of these normal cell lines were immortalized and the impact of immortalization on radiation response is complex and context-dependent, and further research is needed to fully understand the underlying mechanisms.” Lines 294 to 301
Why is the BRCA1 construction also TP53-/- ?
Authors: The inactivation of BRCA1, a critical gene involved in DNA repair, has been shown to trigger a robust p53-dependent cell cycle arrest response, making these cells unsuitable for clonal selection and radiosensitivity evaluation. To overcome this arrest, BRCA1 was inactivated in cells that are null for p53. This approach was employed to bypass the cell cycle arrest response triggered by BRCA1 loss and is a technique widely used and well described in the literature.
Minor comment:
“Figure 1: The points and values do not appear on the figure”
Authors: We apologize for the lack of information in the figure, which appears to not display properly in some software and operating systems. An updated version has been uploaded which will hopefully address this issue.
Line 62 :comment, in the list of the disadvantages of clonogenic assay, do not forgot to cite the fact that some cell lines have a very poor efficiency of clone formation that prevents their use for radiosensitivity measurement.
Authors: Done as recommended: “Furthermore, the clonogenic assay has limitations, including the extended time required for colony growth (typically 7 to 10 days or longer), and the low efficiency of clone formation in certain cell lines, which can make this assay unsuitable for such cells” line 62 to 64.
Figure 3 H : These results should be commented. Number of cell line in common, differences between survival measurements methods etc…
Authors: Done as recommended: “Indeed, our findings show remarkable concordance with previously published results from a CRISPR screen studying radiation response in RPE-1 p53-/- cells, despite employing different methods to assess the impact of DNA repair deficiencies on radiosensitivity. Specifically, while we investigated clonogenic survival, the screen study quantified raw numbers of cells with loss of particular genes (15). This not only underscores the reproducibility of different approaches, but also increases confidence in the robustness of our data presented here. We observed a near-perfect correlation among various knockout models, with the exception of BRCA1 null cells (R2=0.98, p=0.002). This discrepancy in BRCA1 data could potentially be attributed to the limitations of CRISPR dropout screens in genes that are essential for cell proliferation, as well as genes with multiple cellular functions, as has been previously reported in BRCA1 (17). Nonetheless, despite this discrepancy, our results align closely with previously reported data and strengthen the overall reliability of our findings.” Lines 258 to 270.
Figure 7 : the color scale is entirely white!
Authors: We apologize for the lack of information in the figure, which appears to result from the same issue as Figure 1. An updated version has been uploaded which will hopefully address this issue.
Table 1 : there is an error in the cell cycle counts of RPE1-DCLRE1C (>100%). This cell line is not commented in the results
Authors: We apologize for incorrect value in the percentages of cells in G2/M for this cell model that should have been 23.9 and not 28.9. This has been corrected in the table. This cell line is reported in the text as Artemis null cell lines, and we have clarified that DCLRE1C is the gene that encodes the Artemis protein.
Results §2 line 119-124 : the use of DSB is inexact and should be defined as it is 53BP1 foci that were measured and not directly DSBs. Similarly the terms of residual DSBs and induced DSBs should be more precisely described (53BP1 foci at 24h and 1h?).
Authors: We apologize if this was not clear in the first version on the manuscript. In the updated version it is clearly said that 53BP1 foci were used as an indirect measure of DSBs.
Figure 5. Since residual DSBs (53BP1foci at 24h) seem to be the marlker with the best correlation to survival in normal cells why are the induced DSBs presented and discussed and not residual DSBs? Precise the time of the 53BP1 measurement in the legend of the figure
Authors: We apologize if this was not clear in the first version on the manuscript, residual damage (53BP1 foci 24h after exposure) is the marker with the best correlation to survival in the CRISPR subgroup of cells, but it performs poorly in all other subgroups and the complete dataset, including the subgroup of Normal cells. Induced DSBs (53BP1 foci 1hour after exposure) is the marker with best R2 value in the subgroup of Prostate cancer cell lines but fails to be statistically significant. Thought the discussion in the manuscript we have discussed both facts and highlighted previous publications regarding different metrics. As recommended, time of the 53BP1 measurements were added to the figure legend.
Reviewer 2 Report
The manuscript “Characterization of intrinsic radiation sensitivity in a diverse panel of normal, cancerous and CRISPR-modified cell lines” by Guerra Liberal and McMahon, has investigated the sensitivity of a number of human cell lines to ionizing radiation (X rays) in order to provide a data set for model development and for defining potential determinants of radiosensitivity.
For this purpose, they have investigated induction and repair of DNA double strand breaks (DSBs), cell cycle distribution, ploidy, and clonogenic survival following X-ray irradiation.
These parameters including the Mean Inactivation Dose (MID), were determined for a number of cell lines (normal and tumoral origin), including some cell lines derived from RPE-1 cells and characterized by genetic defects in various DNA repair factors, obtained by CRISPR-Cas9 technology.
Although the aim of this manuscript is potentially interesting and important, several aspects were not considered in this study, the most important of which is the comparison between normal and tumor cell lines from different origin.
For instance, in Figure 4 the Survival curves and MDI of lung and prostate cancer cell lines, are compared with normal cell lines from different origin, while each tumor cell lines should be compared only with normal cells from the same tissue type (e.g., PC-3, DU145 etc., should be compared only with PNT1A and PTN2 cells). For instance, the normal MCF10A cells from breast tissue should be compared with tumor cells of the same origin (e.g., MCF7 or others). The same holds true for Figure 5, and all Tables in which normal cell parameters have been reported all together.
As far as the use of CRISPR-Cas9 modified cell lines, no evidence (e.g., Western blots) documenting the successful deletion or modification of the target genes, are presented, although a section in the Materials and methods describes the validation. These data should be added as a Supplementary Figure.
The same should be done for some microscope images showing the staining of immunofluorescence assays used to derive the data of 53BP1, or gamma-H2AX foci. In addition, since a true DNA repair assay assessing the removal of DSB (e.g., the Comet or equivalent assay) has been not performed, and only the disappearance of gamma-H2AX foci, which is an indirect evaluation, has been presented, the Authors should either present this type of evaluation, or indicate properly the parameter used.
Finally, the Authors state in the Conclusions that “…Residual radiation-induced DSBs …. fail to predict survival in a group of cells with a more complex genetic profile”. Indeed, this sentence outline the main problem of comparing data obtained from cells of different tissue origin.
A more specific sentence should be written after the comparison is done following this principle.
Other points:
Figure 1 is not complete, data are missing. Please, provide.
Figure 7: the color code following the graduation of R2 is not shown.
Author Response
Reviewer 2:
Reviewer: “The manuscript “Characterization of intrinsic radiation sensitivity in a diverse panel of normal, cancerous and CRISPR-modified cell lines” by Guerra Liberal and McMahon, has investigated the sensitivity of a number of human cell lines to ionizing radiation (X rays) in order to provide a data set for model development and for defining potential determinants of radiosensitivity.
For this purpose, they have investigated induction and repair of DNA double strand breaks (DSBs), cell cycle distribution, ploidy, and clonogenic survival following X-ray irradiation.
These parameters including the Mean Inactivation Dose (MID), were determined for a number of cell lines (normal and tumoral origin), including some cell lines derived from RPE-1 cells and characterized by genetic defects in various DNA repair factors, obtained by CRISPR-Cas9 technology.
Although the aim of this manuscript is potentially interesting and important, several aspects were not considered in this study, the most important of which is the comparison between normal and tumor cell lines from different origin.
For instance, in Figure 4 the Survival curves and MDI of lung and prostate cancer cell lines, are compared with normal cell lines from different origin, while each tumor cell lines should be compared only with normal cells from the same tissue type (e.g., PC-3, DU145 etc., should be compared only with PNT1A and PTN2 cells). For instance, the normal MCF10A cells from breast tissue should be compared with tumor cells of the same origin (e.g., MCF7 or others). The same holds true for Figure 5, and all Tables in which normal cell parameters have been reported all together.”
Authors: We sincerely thank the reviewer for their thoughtful evaluation of our manuscript and appreciate the reviewer's feedback.
We would first like to clarify a point about the comparisons made within this work. Except the final pooled analyses across all lines considered, we do not make any direct computational comparisons between cancer cells and normal cells within this work, with the ‘prostate’ and ‘lung’ groups only including cancer cell lines, with no normal cells. Thus all of these comparisons are only amongst tumours derived from the same site, and almost all from the same type of tissue (all prostate lines are prostate carcinoma, all but one lung line are adenocarcinoma). As our work has highlighted the significant variability in response between normal tissues, even from the same site, there are insufficient normal cell lines in many sites for us to be confident we have sampled the true variability, which makes subsequent comparisons difficult.
We acknowledge that comparing normal cell lines with tumor cell lines from the same tissue origin could provide valuable mechanistic insights. However, our primary aim in this study was to investigate the sensitivity of cancer cells to radiation, with a focus on identifying potential markers for radiosensitivity. As such, we chose to compare among different cell lines from the same cancers, focusing on prostate and lung tissues, which are clinically relevant in the context of radiation therapy for cancer treatment. Markers which would distinguish between cancers of the same type would be most directly applicable for clinical translation.
We understand that comparing normal cell lines with tumor cell lines from the same tissue origin could provide additional insights from a mechanistic standpoint, we have noted that there can be significant variability among normal cell lines, which may limit the interpretability of such comparisons. Ideally such a comparison would exploit matched ‘pairs’ of normal and cancerous tissue from the same patient to account for this as much as possible, but such resources are unfortunately not readily available.
Additionally, our study was designed to investigate the complexity of cancer cells and the impact of various genetic alterations on radiation response. Therefore, we opted to include normal cell lines from different tissue origins as a reference and we believe that our approach of comparing tumor cell lines with each other and normal cells, despite originating from different tissues, is relevant to the clinical question of radiation sensitivity in cancer treatment.
Reviewer: “As far as the use of CRISPR-Cas9 modified cell lines, no evidence (e.g., Western blots) documenting the successful deletion or modification of the target genes, are presented, although a section in the Materials and methods describes the validation. These data should be added as a Supplementary Figure.”
Authors: We apologize for the lack of information regarding Western Blot validation, this was added in the updated version of the manuscript as supplementary figure 3.
Reviewer:The same should be done for some microscope images showing the staining of immunofluorescence assays used to derive the data of 53BP1, or gamma-H2AX foci. In addition, since a true DNA repair assay assessing the removal of DSB (e.g., the Comet or equivalent assay) has been not performed, and only the disappearance of gamma-H2AX foci, which is an indirect evaluation, has been presented, the Authors should either present this type of evaluation, or indicate properly the parameter used.
Authors: We apologize for the lack of information regarding 53BP1 analysis, representative figures of 4 different cells RPE-1 WT, RPE-1 p53 null, RPE-1 LIG4 null and RPE-1 p53/BRCA1 null at different exposure times 1h, 24 h or control samples were added in the updated version of the manuscript as supplementary figure 4. The text has been updated in several places to clarify that 53BP1 foci are being used as an indirect marker of DNA damage and repair (e.g. line 122).
Reviewer: Finally, the Authors state in the Conclusions that “…Residual radiation-induced DSBs …. fail to predict survival in a group of cells with a more complex genetic profile”. Indeed, this sentence outline the main problem of comparing data obtained from cells of different tissue origin.
A more specific sentence should be written after the comparison is done following this principle.
Authors: As noted above, within the main prostate and lung cancer comparison groups, all cells which were compared were from the same site and in almost all cases the same tissue type, and the performance of the relevant biomarkers remained poor even within these groups. This may be a confounding factor in the normal tissue analysis, and some text discussing potential reasons for the variation in normal tissues has been added at line 294, including the reviewer’s point about tissue of origin.
Other points:
Figure 1 is not complete, data are missing. Please, provide.
Authors: We apologize for the lack of information in the figure, which appears to not display properly in some software and operating systems. An updated version has been uploaded which will hopefully address this issue.
Figure 7: the color code following the graduation of R2 is not shown.
Authors: We apologize for the lack of information in the figure, which appears to result from the same issue as Figure 1. An updated version has been uploaded which will
Reviewer 3 Report
The goal of this research was to identify a more rapid predictive assay of radiosensitivity of cancer cells in patients receiving radiotherapy.
This paper provides comparisons for radiosensitivity based on cell survival, proliferation and DNA repair end-points in 27 different cell lines. The authors first studied DNA repair related genes in a number of CRISPR modified cell lines from the RPE-1 (normal) cell line. They then studied commonly used cell lines from normal tissues, prostate cancers and lung cancers. The authors had hoped to find markers that could predict radiation sensitivity in tumours based on the accumulated information from these cell lines. They observed that with only a few exceptions, many of the cell lines behaved in a similar fashion regardless of their genetic make-up related to DNA repair. The literature is littered with radiation studies on different normal and cancer cell lines with varying responses for different radiation damage metrics and using different doses. This paper makes a valuable contribution in that it systematically compares a large number of cell lines for a number of radiation doses, for several end-points. It is well written, easy to follow, carefully executed with appropriate controls and number of replicates, and the figures and tables are clear. It also presents a substantial amount of work. Current dogma is focussed on the importance of DNA repair in radiosensitivity of cancer and normal cells. This paper indicates that DNA repair is a good predictor in a few specific circumstances, but not so useful for the complex heterogeneity of cancer cells likely to be encountered in patients. The authors suggest that there should be more focus on other pathways involved in proliferation and metabolism.
General Comments:
Some justification should be given for why RPE-1 was chosen for the CRISPR studies? This is a normal cell line but it comes from retina. Subsequently all the comparisons are made with non-retina normal lines (largely from prostate) or prostate cancer and lung cancer cell lines. Why not perform CRISPR on e.g. the normal prostate cell line RWPE1 or normal lung MRC5? Interestingly, retina is a highly metabolic tissue.
The discussion of the data is informative but finishes rather abruptly by suggesting that proliferation (some evidence from the current study) and metabolism should be studied in relation to radiosensitivity rather than DNA repair. This is a perfectly feasible, and as a result of this paper a justified, suggestion but could do with more substantiation in the discussion, especially for metabolism and the potential importance of antioxidant pathways.
Minor comments:
Figure 1 has not come out in a readable fashion on the pdf that I had access to. It has axes with one small horizontal line for A459 and nothing else? I can imagine and accept that there is great variation amongst them but could not assess.
Line 101: Should end by referring to Figure 2D.
Line 224: preforms should read as perform
Line 225: should read percentage of cells, not percentage of cell
Figure 7; line 234. Should read - likely due to their
Line 244: should read – this is in agreement
Line 317 and 318. Replace the word believe with something more scientific, e.g. hypothesise.
Line 337: should read – were generated
Line 345: check superscript
Section 4.3. line 352. Title does not make sense. Western Blot got Knock-out validation. Should got be and?
Author Response
Reviewer 3:
Reviewer: “The goal of this research was to identify a more rapid predictive assay of radiosensitivity of cancer cells in patients receiving radiotherapy. This paper provides comparisons for radiosensitivity based on cell survival, proliferation and DNA repair end-points in 27 different cell lines. The authors first studied DNA repair related genes in a number of CRISPR modified cell lines from the RPE-1 (normal) cell line. They then studied commonly used cell lines from normal tissues, prostate cancers and lung cancers. The authors had hoped to find markers that could predict radiation sensitivity in tumours based on the accumulated information from these cell lines. They observed that with only a few exceptions, many of the cell lines behaved in a similar fashion regardless of their genetic make-up related to DNA repair. The literature is littered with radiation studies on different normal and cancer cell lines with varying responses for different radiation damage metrics and using different doses. This paper makes a valuable contribution in that it systematically compares a large number of cell lines for a number of radiation doses, for several end-points. It is well written, easy to follow, carefully executed with appropriate controls and number of replicates, and the figures and tables are clear. It also presents a substantial amount of work. Current dogma is focussed on the importance of DNA repair in radiosensitivity of cancer and normal cells. This paper indicates that DNA repair is a good predictor in a few specific circumstances, but not so useful for the complex heterogeneity of cancer cells likely to be encountered in patients. The authors suggest that there should be more focus on other pathways involved in proliferation and metabolism.”
Authors: We appreciate the reviewer's feedback and recognition of the contribution of our study in systematically comparing a large number of cell lines for radiation sensitivity We appreciate the reviewer's positive feedback on the quality of our study, including the careful execution with appropriate controls and replicates, clear figures and tables, and well-written presentation of the results. We will ensure that the implications of our findings regarding the limitations of using DNA repair as a predictor of radiation sensitivity in the complex heterogeneity of cancer cells are clearly addressed in the manuscript. We also acknowledge the importance of exploring other pathways involved in proliferation and metabolism as potential markers for radiation sensitivity in future studies. and we are grateful for Reviewer’s constructive comments that have helped improve our paper.
General Comments:
Reviewer: “Some justification should be given for why RPE-1 was chosen for the CRISPR studies? This is a normal cell line but it comes from retina. Subsequently all the comparisons are made with non-retina normal lines (largely from prostate) or prostate cancer and lung cancer cell lines. Why not perform CRISPR on e.g. the normal prostate cell line RWPE1 or normal lung MRC5? Interestingly, retina is a highly metabolic tissue.”
Authors: Following the Reviewer’s recommendation we have now justified the selection of RPE-1 for CRISPR constructions and cell selection process as the following: ““The RPE-1 cells were selected for CRISPR-Cas9 studies due to their stable epithelial cell characteristics, with minimal genetic modifications, which closely mimic normal tissue biology. Additionally, its widespread usage in previous CRISPR studies provides a robust foundation of data for comparison, confirming its suitability as a reliable and well-established cellular model. The process of selecting tumoral and normal human cell lines in this study was carried out based on factors such as cell availability, previously described levels of radiosensitivity and their widespread use in radiation studies. Lung and prostate cell lines were selected in this initial comparison as radiotherapy is frequently used as a primary treatment in these cancers.” This can be found in line 385 to 394 of the manuscript.
While RWPE1 and MRC5 are commonly used normal cell lines for prostate and lung tissues, respectively, they may have limitations for CRISPR manipulation and subsequent radiobiological studies: MRC5 is a non-immortalized fibroblast cell line, with a limited number of potential cellular divisions which makes it impossible to produce a stable cell line for future work. It also has a poor clonogenic efficiency. For RWPE1, despite being a normal epithelial cell line they have been immortalized using HPV18, which leads to elevated levels of background damage, and potential alteration of functions of p53 and other regulatory genes, which could potentially affect the outcome and interpretation of those knockout studies.
Reviewer: “The discussion of the data is informative but finishes rather abruptly by suggesting that proliferation (some evidence from the current study) and metabolism should be studied in relation to radiosensitivity rather than DNA repair. This is a perfectly feasible, and as a result of this paper a justified, suggestion but could do with more substantiation in the discussion, especially for metabolism and the potential importance of antioxidant pathways.”
Authors: Following the Reviewer’s recommendation we have now extended the discussion regarding possible impacts of cellular metabolism, proliferation and antioxidant pathways as the following: “Furthermore, metabolic pathways also control the production of energy and essential molecules needed for cell survival and repair. Understanding how radiation affects cellular metabolism can provide insights into the mechanisms of radiation response and may lead to strategies for enhancing radiation efficacy. Disruption of cellular metabolism with mannose has been shown to result in impaired cancer cell growth and survival (25). This effect is thought to be mediated through multiple mechanisms, including the disruption of glycolysis, leading to a decreased energy production, modulation of signaling pathways involved in cancer progression, such as the PI3K/Akt pathway and enhancement of anti-cancer immune response (26). Similarly, radiation exposure increases the levels of reactive oxygen species (ROS) and other damaging molecules, overwhelming the cellular antioxidant defense mechanisms. Subsequently, this can cause oxidative stress, further DNA damage and cell death. Antioxidant pathways, such as the glutathione system, superoxide dismutase, and catalase, help to neutralize ROS and prevent oxidative damage (27). Modulating these antioxidant pathways may enhance the cellular antioxidant capacity and reduce radiation-induced damage (28). In summary a better understanding of the interplay between cellular metabolism, proliferation, antioxidant pathways, and radiation response is essential for elucidating the mechanisms of radiation response and may lead to the development of strategies to increase radiation efficacy in cancer treatments.” Lines 353 to 372
Minor comments:
Reviewer: “Figure 1 has not come out in a readable fashion on the pdf that I had access to. It has axes with one small horizontal line for A459 and nothing else? I can imagine and accept that there is great variation amongst them but could not assess.”
Authors: We apologize for the lack of information in the figure, which appears to not display properly in some software and operating systems. An updated version has been uploaded which will hopefully address this issue.
Reviewer: “Line 101: Should end by referring to Figure 2D.”
Authors: Done as recommended
Reviewer: “Line 224: preforms should read as perform”
Authors: Done as recommended
Reviewer: “Line 225: should read percentage of cells, not percentage of cell”
Authors: Done as recommended
Reviewer: “Figure 7; line 234. Should read - likely due to their”
Authors: Done as recommended
Reviewer: “Line 244: should read – this is in agreement”
Authors: Done as recommended
Reviewer:”Line 317 and 318. Replace the word believe with something more scientific, e.g. hypothesise.”
Authors: Done as recommended
Reviewer: “Line 337: should read – were generated “
Authors: Done as recommended
Reviewer: “Line 345: check superscript”
Authors: Done as recommended
Reviewer: “Section 4.3. line 352. Title does not make sense. Western Blot got Knock-out validation. Should got be and?”
Authors: Section title was changed to Knock-out validation by Western Blot
Round 2
Reviewer 2 Report
The Authors have answered to my issues, and explained their approach. The manuscript can be accepted for publication.